

# Research artifacts and citations in computer systems papers

Eitan Frachtenberg

Computer Science, Reed College, Portland, OR, United States of America

## ABSTRACT

Research in computer systems often involves the engineering, implementation, and measurement of complex systems software and data. The availability of these artifacts is critical to the reproducibility and replicability of the research results, because system software often embodies numerous implicit assumptions and parameters that are not fully documented in the research article itself. Artifact availability has also been previously associated with higher paper impact, as measured by citations counts. And yet, the sharing of research artifacts is still not as common as warranted by its importance. The primary goal of this study is to provide an exploratory statistical analysis of the artifact-sharing rates and associated factors in the research field of computer systems. To this end, we explore a cross-sectional dataset of papers from 56 contemporaneous systems conferences. In addition to extensive data on the conferences, papers, and authors, this analyze dataset includes data on the release, ongoing availability, badging, and locations of research artifacts. We combine this manually curated dataset with citation counts to evaluate the relationships between different artifact properties and citation metrics. Additionally, we revisit previous observations from other fields on the relationships between artifact properties and various other characteristics of papers, authors, and venue and apply them to this field. The overall rate of artifact sharing we find in this dataset is approximately 30%, although it varies significantly with paper, author, and conference factors, and it is closer to 43% for conferences that actively evaluated artifact sharing. Approximately 20% of all shared artifacts are no longer accessible four years after publications, predominately when hosted on personal and academic websites. Our main finding is that papers with shared artifacts averaged approximately 75% more citations than papers with none. Even after controlling for numerous confounding covariates, the release of an artifact appears to increase the citations of a systems paper by some 34%. This metric is further boosted by the open availability of the paper's text.

# INTRODUCTION

Many scientific experimental results cannot be successfully repeated or reproduced, leading to the so-called "reproducibility crisis" (*Baker, 2016b*; *Van Noorden, 2015*). An experimental result is not fully established unless it can be independently reproduced (*Stodden, 2008*), and an important step towards this goal is the sharing of artifacts associated with the work, including computer code (*ACM, 2020*; *Collberg & Proebsting, 2016*; *Fehr et al., 2016*). The availability of experimental artifacts is not only

Corresponding author
Eitan Frachtenberg,
etc_26@yahoo.com

crucial for reproducibility, but it also directly contributes to the transparency, reusability, and credibility of the work (*Feitelson, 2015*). Artifacts additionally play an important role in drive toward open science, which has gained substantial momentum in computer science (CS) (*Heumüller et al., 2020*).

Given the central role of research artifacts in reproducibility, it is not surprising to find major initiatives to increase artifact sharing and evaluation. As Childers and Chrysanthis wrote in 2017:

> Experimental computer science is far from immune [from the reproducibility crisis], although it should be easier for CS than other sciences, given the emphasis on experimental artifacts, such as source code, data sets, workflows, parameters, etc. The data management community pioneered methods at ACM SIGMOD 2007 and 2008 to encourage and incentivize authors to improve their software development and experimental practices. Now, after 10 years, the broader CS community has started to adopt Artifact Evaluation (AE) to review artifacts along with papers (*Childers & Chrysanthis, 2017*).

Unfortunately, the sharing and evaluation of artifacts are still not as commonplace as warranted by their importance. One challenge in addressing this topic is that definitions and expectations for research artifacts are not always clear. The Association of Computing Machinery (ACM) defines a paper's artifact as follows:

> By "artifact" we mean a digital object that was either created by the authors to be used as part of the study or generated by the experiment itself. For example, artifacts can be software systems, scripts used to run experiments, input datasets, raw data collected in the experiment, or scripts used to analyze results (*ACM, 2020*).

This paper aims to shed some light on artifact sharing in one particular field of CS, namely *computer systems* (or "systems" for short). Systems is a large research field with numerous applications, used by some of the largest technology companies in the world. For the purpose of this study, we define systems as the study and engineering of concrete computing systems, which includes research topics such as: operating systems, computer architectures, data storage and management, compilers, parallel and distributed computing, and computer networks.

The study of these topics often involves the implementation, modification, and measurement of *system software* itself. System software is software that is not on its own a user-facing application, but rather software that manages system resources or facilitates development for the actual applications, such as compilers, operating system components, databases, and middleware. System software can be fairly complex and tightly coupled to the system it is designed to run on.

Research artifacts, and especially software artifacts, are therefore paramount to the evaluation and reproduction of systems research. Whereas in other fields of science—or even CS—research results can often be replicated from the original equations or datasets, systems software can embody countless unstated assumptions in the code and parameters. The significance is that reproducing research findings by recreating its artifacts from the

terse descriptions in a paper is often unfeasible, rendering software artifacts all that more important.

The hypothesis of this paper is therefore that because of its importance to reproducibility, artifacts sharing in systems significantly increases a paper's influence, as measured by citations. Citations are not only a widely used metric of impact for papers and researchers but also an indirect measure of the work's quality and usefulness, which presumably are both helped by the availability of artifacts. Citations may also stand in as proxy metrics for the transparency, reusability, reproducibility, and credibility of papers—if we assume that any of these qualities encourage subsequent researchers to cite the original work. The main observational goal of this paper is to evaluate the quantitative association between artifacts availability and citations in the research field of computer systems. An additional goal of this study is an exploratory data analysis of associated factors to identify relationships between the availability of research artifacts in systems research papers and other statistical properties of these papers.

## Study design and main findings

To evaluate our main hypothesis, this study uses an observational, cross-sectional approach, analyzing 2,439 papers from a large subset of leading systems conferences. The study population comes from a hand-curated collection of 56 peer-reviewed systems and related conferences from a single publication year (2017). Among other characteristics, it includes manually collected data on artifact availability and paper citation counts 3.5 years from publication, as detailed in the next section. By comparing the post-hoc citations of papers with released artifacts to those with none, we find that we can reject the null hypothesis that artifact availability does not impact paper citations. Even after controlling for demographic, paper, and conference factors using a multilevel mixed-effects model, papers with artifacts still receive 34% more citations on average than papers without.

Our expansive dataset also offers the opportunity for a descriptive and correlational study of the following additional questions. These questions are ancillary to the main research question of the relationship between artifacts and citations. Nevertheless, they may interest the reader and provide a fuller context and quantitative understanding of the state of artifact sharing in the field, and are provided as secondary contributions. These questions, and a short answer to each, are listed here and are elaborated in the results section:

1. What is the ratio of papers in systems for which artifacts are available? (Approximately 30%.)
2. How many of these artifacts are actually linked from the paper? (Approximately 80%.)
3. How many of these artifacts have expired since publication? What characterizes these artifacts? (Approximately 13% can no longer be found, mostly from academic and personal host pages.)
4. What are the per-conference factors and differences that affect the ratio of artifact sharing? (The most influential factor appears to be an artifact evaluation process. Approximately 57% of papers in these six conferences shared artifacts.)

5. Does conference prestige affect artifact availability? (Papers that release artifacts tend to appear in more competitive conferences.)
6. What is the relationship between artifact accessibility and paper accessibility? (Papers with shared artifacts are also more likely to have an eprint version freely available online, and sooner than non-artifact papers.)
7. What is the relationship between artifact accessibility and paper awards? (Approximately 39% of papers with awards shared artifacts *vs.* 27% in the rest.)
8. Are there any textual properties of the paper that can predict artifact availability? (Papers that share artifacts tend to be longer and incorporate a computer system moniker in their titles.)

As a final contribution, this study provides a rich dataset of papers (*Frachtenberg, 2021*), tagged with varied metadata from multiple sources, including for the first time artifact properties (described next). Since comprehensive data on papers with artifacts is not always readily available, owing to the significant manual data collection involved, this dataset can serve as the basis of additional studies.

The rest of this paper is organized as follows. The next section presents in detail the dataset, methodology, and limitations of our study. An extensive set of descriptive and explanatory statistics is presented in the results section and then used to build a mixed-effects multilevel regression model for citation count. The discussion section presents potential implications from these findings, as well as potential threats to the validity of this analysis. These findings are then placed in historical and cross-disciplinary context in the related-work section. Finally, the concluding section summarizes the main findings of this study and suggests some future research directions.

## DATA AND METHODS

The most time-consuming aspect of this study was the collection and cleaning of the data. This section describes the data selection and cleaning process for paper, artifact, and citation data.

The primary dataset we analyze comes from a hand-curated collection of 56 peer-reviewed systems and related conferences from a single publication year (2017), to reduce time-related variance. Conference papers were preferred over journal articles because in CS, and in particular, in its more applied fields such as systems, original scientific results are typically first published in peer-reviewed conferences (*Patterson, Snyder & Ullman, 1999*; *Patterson, 2004*):health, and then possibly in archival journals, sometimes years later (*Vrettas & Sanderson, 2015*). These conferences were selected to represent a large cross-section of the field, with different sizes, competitiveness, and subfields (Table 1). Such choices are necessarily subjective, based on the author's experience in the field. But they are aspirationally both wide enough to represent the field well and focused enough to distinguish it from the rest of CS.

A few of these conferences, such as MobiCom and SLE, specifically encouraged artifacts in their call-for-papers or websites. Four conferences—SC, OOPSLA, PLDI, and SLE—archived their artifacts in the ACM's digital library. In addition to general

**Table 1  System conferences, including start date, number of published papers, total number of named authors, and acceptance rate.**

| Conference | Date | Papers | Authors | Acceptance | Conference | Date | Papers | Authors | Acceptance |
|---|---|---|---|---|---|---|---|---|---|
| ICDM | 2017-11-19 | 72 | 269 | 0.09 | PACT | 2017-09-11 | 25 | 89 | 0.23 |
| KDD | 2017-08-15 | 64 | 237 | 0.09 | SPAA | 2017-07-24 | 31 | 84 | 0.24 |
| SIGMETRICS | 2017-06-05 | 27 | 101 | 0.13 | MASCOTS | 2017-09-20 | 20 | 75 | 0.24 |
| SIGCOMM | 2017-08-21 | 36 | 216 | 0.14 | CCGrid | 2017-05-14 | 72 | 296 | 0.25 |
| SP | 2017-05-22 | 60 | 287 | 0.14 | PODC | 2017-07-25 | 38 | 101 | 0.25 |
| PLDI | 2017-06-18 | 47 | 173 | 0.15 | CLOUD | 2017-06-25 | 29 | 110 | 0.26 |
| NDSS | 2017-02-26 | 68 | 327 | 0.16 | Middleware | 2017-12-11 | 20 | 91 | 0.26 |
| NSDI | 2017-03-27 | 42 | 203 | 0.16 | EuroPar | 2017-08-30 | 50 | 179 | 0.28 |
| IMC | 2017-11-01 | 28 | 124 | 0.16 | PODS | 2017-05-14 | 29 | 91 | 0.29 |
| ISCA | 2017-06-24 | 54 | 295 | 0.17 | ICPP | 2017-08-14 | 60 | 234 | 0.29 |
| SOSP | 2017-10-29 | 39 | 217 | 0.17 | ISPASS | 2017-04-24 | 24 | 98 | 0.30 |
| ASPLOS | 2017-04-08 | 56 | 247 | 0.18 | Cluster | 2017-09-05 | 65 | 273 | 0.30 |
| CCS | 2017-10-31 | 151 | 589 | 0.18 | OOPSLA | 2017-10-25 | 66 | 232 | 0.30 |
| HPDC | 2017-06-28 | 19 | 76 | 0.19 | HotOS | 2017-05-07 | 29 | 112 | 0.31 |
| MICRO | 2017-10-16 | 61 | 306 | 0.19 | ISC | 2017-06-18 | 22 | 99 | 0.33 |
| MobiCom | 2017-10-17 | 35 | 164 | 0.19 | HotCloud | 2017-07-10 | 19 | 64 | 0.33 |
| ICAC | 2017-07-18 | 14 | 46 | 0.19 | HotI | 2017-08-28 | 13 | 44 | 0.33 |
| SC | 2017-11-13 | 61 | 325 | 0.19 | SYSTOR | 2017-05-22 | 16 | 64 | 0.34 |
| CoNEXT | 2017-12-13 | 32 | 145 | 0.19 | ICPE | 2017-04-22 | 29 | 102 | 0.35 |
| SIGMOD | 2017-05-14 | 96 | 335 | 0.20 | HotStorage | 2017-07-10 | 21 | 94 | 0.36 |
| PPoPP | 2017-02-04 | 29 | 122 | 0.22 | IISWC | 2017-10-02 | 31 | 121 | 0.37 |
| HPCA | 2017-02-04 | 50 | 215 | 0.22 | CIDR | 2017-01-08 | 32 | 213 | 0.41 |
| EuroSys | 2017-04-23 | 41 | 169 | 0.22 | VEE | 2017-04-09 | 18 | 85 | 0.42 |
| ATC | 2017-07-12 | 60 | 279 | 0.22 | SLE | 2017-10-23 | 24 | 68 | 0.42 |
| HiPC | 2017-12-18 | 41 | 168 | 0.22 | HPCC | 2017-12-18 | 77 | 287 | 0.44 |
| SIGIR | 2017-08-07 | 78 | 264 | 0.22 | HCW | 2017-05-29 | 7 | 27 | 0.47 |
| FAST | 2017-02-27 | 27 | 119 | 0.23 | SOCC | 2017-09-25 | 45 | 195 | Unknown |
| IPDPS | 2017-05-29 | 116 | 447 | 0.23 | IGSC | 2017-10-23 | 23 | 83 | Unknown |

encouragement and archival, six conferences specifically offered to evaluate artifacts by a technical committee: OOPSLA, PACT, PLDI, PPoPP, SC, and SLE.

For each conference, we gathered various statistics from its web page, proceedings, or directly from its chairs. We also collected historical conference metrics from the websites of the ACM, the Institute of Electrical and Electronics Engineers (IEEE), and Google Scholar (GS), including past citations, age, and total publications, and downloaded all papers in PDF format. The dataset includes extensive data on the authors and the textual properties of the papers, and the relevant features are discussed in the next section.

We are also interested in measuring the post-hoc impact of each paper, as approximated by its number of citations. Citation metrics typically lag publication by a few months or years, allowing for the original papers to be discovered, read, cited, and then the citations themselves published and recorded. The time duration since these papers had been published, approximately 3.5 years, permits the analysis of their short-to-medium term impact in terms of citations. In practice, this duration is long enough that only 46 papers (1.89%) have gathered no citations yet.

For this study, the most critical piece of information on these papers is their artifacts. Unfortunately, most papers included no standardized metadata with artifact information, so it had to be collected manually from various sources, as detailed next.

The only existing form of standardized artifact metadata was for the subset of conferences organized by the ACM with artifact badge initiatives. In the proceedings page in the ACM's digital library of these conferences, special badges denote which papers made artifacts available, and which papers had artifacts evaluated (for conferences that supported either badge). In addition, the ACM digital library also serves as a repository for the artifacts, and all of these ACM papers included a link back to the appropriate web page with the artifact.

Unfortunately, most papers in this dataset were not published by the ACM or had no artifact badges. In the absence of artifact metadata or an automated way to extract artifact data, these papers required a manual scanning of the PDF text of every paper in order to identify such links. When skimming these papers, several search terms were used to assist in identifying artifacts, namely: "github", "gitlab", "bitbucket", "sourceforge", and "zenodo" for repositories; variants of "available", "open source", and "download" for links; and variations of "artifact", "reproducibility", and "will release" for indirect references. Some papers make no mention of artifacts in the text, but we can still discover associated artifacts online by searching github.com for author names, paper titles, and especially unique monikers used in the paper to identify their software.

We also recorded for each paper: whether the paper had an "artifact available" badge or "artifact evaluated" badge, whether a link to the artifact was included in the text, the actual URL for the artifact, and the latest date that this artifact was still found intact online. All of the searches for these artifacts are recent, so from the last field above we can denote the current status of an artifact as either *extant* or *expired*. From the availability of a URL, we can classify an artifact as *released* or *unreleased* (the latter denoting papers that promised an artifact but no link or repository was found). And from the host domain of the URL we can classify the location of the artifact as either an *Academic* web page, the *ACM* digital library, a *Filesharing* service such as Dropbox or Google, a specialized *Repository* such as github.com, *Other* (including .com and .org web sites), or *NA*.

In all, 722 papers in our dataset (29.6%) had an identifiable or promised artifact, predominantly as software but occasionally as data, configuration, or benchmarking files. Artifacts that had been included in previous papers or written by someone other than the paper's authors were excluded from this count. This statistic only reflects artifact availability, not quality, since evaluating artifact quality is both subjective and time-consuming. It is worth noting, however, that most of the source-code repositories in these artifacts showed

no development activity—commits, forks, or issues—after the publication of their paper, suggesting limited activity for the artifacts alone.

## Data-collection procedure

The following list summarizes the data-collection process for reproducibility purposes.

1. Visit the website and proceedings of each conference and record general information about the conference: review policy, open-access, rebuttal policy, acceptance rate, program committee, *etc.*

2. Also from these sources, manually copy the following information for each paper: title, author names, and award status (as noted on the website and in proceedings).

3. Double-check all paper titles and author names by comparing conference website and post-conference proceedings. Also compare titles to GS search results and ensure all papers are (eventually) discovered by GS with the title corrected as necessary. Finally, check the same titles and author names against the Semantic Scholar database and resolve any discrepancies.

4. Download the full text of each paper in PDF format *via* institutional digital library access.

5. Record all papers with artifact badges. These are unique to the ACM conferences in our dataset and are clearly shown both in the ACM digital library and in the PDF copy of such papers.

6. Collect and record GS citation counts for each paper as close to possible to exactly 42 months after the conference's opening day. The dataset includes citation counts for each paper across multiple time points, but the analysis in this paper only uses one data point per paper, closest to the selected duration.

7. Record artifact availability and links for papers. This is likely the most time-consuming and error-prone process in the preparation of the data specific to this study and involves the following steps: Using a search tool on each document ("pdfgrep") on each of the search terms listed above and perusing the results to identify any links or promises to artifacts; skimming or reading papers with negative results to ensure such a link was not accidentally missed; Finally, searching github.com for specific system names if a paper describes one, even if not linked directly from the paper.

## Statistics

For statistical testing, group means were compared pairwise using Welch's two-sample $t$-test; differences between distributions of two categorical variables were tested with $\chi^2$ test; and comparisons between two numeric properties of the same population were evaluated with Pearson's product-moment correlation. All statistical tests are reported with their $p$-values.

## Ethics statement

All of the data for this study was collected from public online sources and therefore did not require the informed consent of the papers' authors.

**Table 2  Class of artifact URLs.** 'NA' locations indicate expired or unreleased URLs.

| Location | Count |
| --- | --- |
| Repository | 477 |
| Academic | 81 |
| Other | 67 |
| ACM | 44 |
| Filesharing | 6 |
| NA | 47 |

## Code and data availability

The complete dataset and metadata are available in the supplementary material, as well as a github repository (*Frachtenberg, 2021*).

## RESULTS

### Descriptive statistics

Before addressing our main research question, we start with a simple characterization of the statistical distributions of artifacts in our dataset. Of the 722 papers with artifacts, we find that about 79.5% included an actual link to the artifact in the text. The ACM digital library marked 88 artifact papers (12.2%) with an "Artifact available" badge, and 89 papers (12.3%) with an "Artifact evaluated" badge. The majority of artifact papers (86.7%) still had their artifacts available for download at the time of this writing. This ratio is somewhat similar to a comparable study that found that 73% of URLs in five open-access (OA) journals were live after five years (*Saberi & Abedi, 2012*). Of the 722 papers that promised artifacts, 47 appear to have never released them. The distribution of the location of the accessible artifacts is shown in Table 2, and is dominated by Github repositories.

Looking at the differences across conferences, Fig. 1 shows the percentage of papers with artifacts per conference, ranging from 0% for ISCA, IGSC, and HCW to OOPSLA's 78.79% (mean: 27.22%, SD: 19.32%). Unsurprisingly, nearly all of the conferences where artifacts were evaluated are prominent in their relatively high artifact rates. Only PACT stands out as a conference that evaluated artifacts but had a lower-than-average overall ratio of papers with artifacts (0.24). The MobiCom conference also shows a distinctly low ratio, 0.09, despite actively encouraging artifacts. It should be noted, however, that many papers in PACT and MobiCom are hardware-related, where artifacts are often unfeasible. The same is true for a number of other conferences with low artifact ratios, such as ISCA, HPCA, and MICRO. Also worth noting is the fact that ACM conferences appear to attract many more artifacts than IEEE conferences, although the reasons likely vary on a conference-by-conference basis.

Another indicator for artifact availability is author affiliation. As observed in other systems papers, industry-affiliated authors typically face more restrictions for sharing artifacts (*Collberg & Proebsting, 2016*), likely because the artifacts hold commercial or competitive ramifications (*Ince, Hatton & Graham-Cumming, 2012*). In our dataset, only

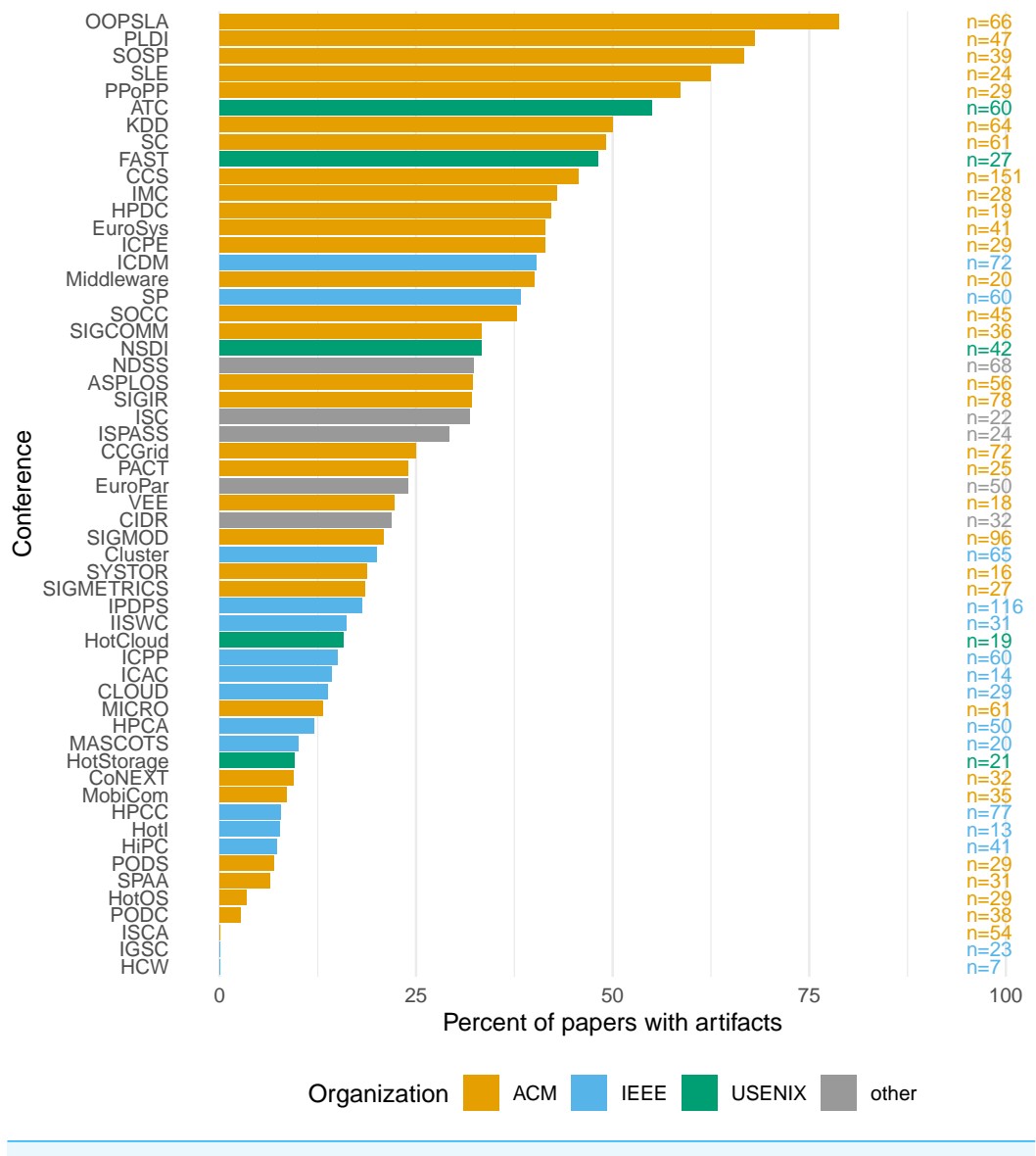

**Figure 1   Papers with artifact by conference.**

19.3% of the 109 papers where all authors had an industry affiliation also released an artifact, compared to 28.1% for the other papers ($\chi^2 = 3.6$, $p = 0.06$).

## Relationships to citations

Turning now to our main research hypothesis, we ask: does the open availability of an artifact affect the citations of a paper in systems? To answer this question, we look at the distribution of citations for each paper 42 months after its conference's opening day, when its proceedings presumably were published[1].

[1] At the time of this writing during summer 2021, the papers from December 2017 had been public for 3.5 years, so this 42-month duration was selected for all papers to normalize the comparison.

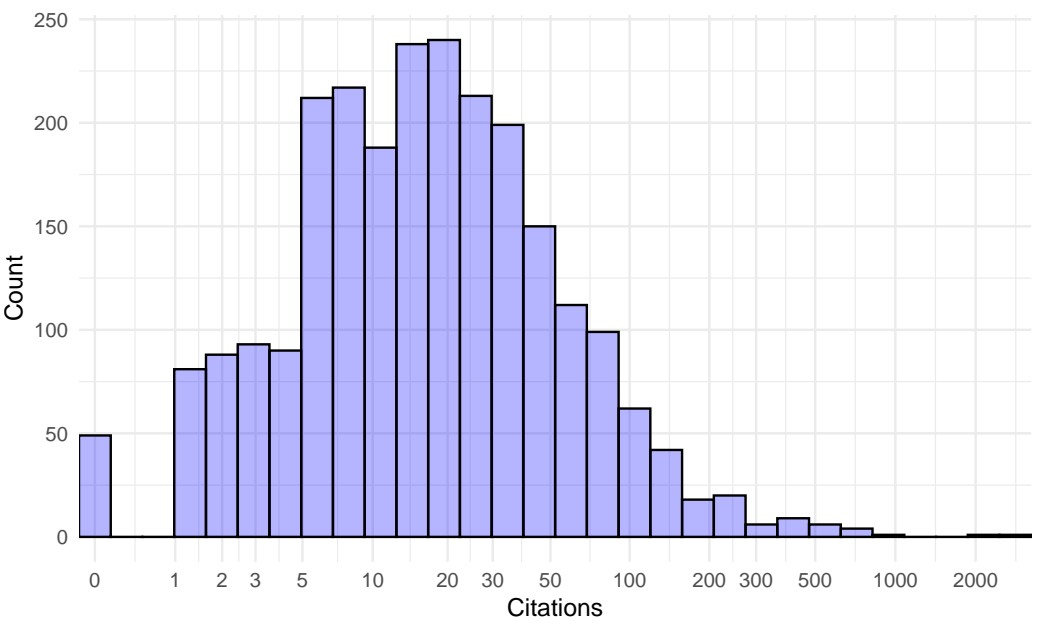

**Figure 2** Distribution of paper citations 42 months after publication (log-scale).

Figure 2 shows the overall paper distribution as a histogram, while Fig. 3 breaks down the distributions of artifact and non-artifact papers as density plots.

Citations range from none at all (49 papers) to about a thousand, with two outlier papers exceeding 2,000 citations (*Carlini & Wagner, 2017*; *Jouppi et al., 2017*). The distributions appear roughly log-normal. The mean citations per paper with artifacts released was 50.7, compared to 29.1 with none ($t = 4.07$, $p < 10^{-4}$). Since the citation distribution is so right-skewed, it makes sense to also compare the median citations with and without artifacts (25 *vs.* 13, $W = 767739$, $p < 10^{-9}$). Both statistics suggest a clear and statistically significant advantage in citations for papers that released an artifact. Likewise, the 675 papers that actually released an artifact garnered more citations than the 47 papers that did promise an artifact that could later not be found ($t = 3.82$, $p < 10^{-3}$), and extant artifacts fared better than expired ones ($t = 4.17$, $p < 10^{-4}$).

In contradistinction, some positive attributes of artifacts were actually associated with fewer citations. For example, the mean citations of the 573 papers with a linked artifact, 47, was much lower than the 71.3 mean for the 102 papers with artifacts we found using a Web search ($t = -2.02$, $p = 0.04$; $W = 22865$, $p < 10^{-3}$). Curiously, the inclusion of a link in the paper, presumably making the artifact more accessible, was associated with fewer citations.

Similarly counter-intuitive, papers that received an "Artifact evaluated" badge fared worse in citations than artifact papers who did not ($t = -3.32$, $p < 0.01$; $W = 11932.5$, $p = 0.03$). Papers who received an "Artifact available" badge did fare a little worse than artifact papers who did not ($t = -1.45$, $p = 0.15$; $W = 26050.5$, $p = 0.56$). These findings

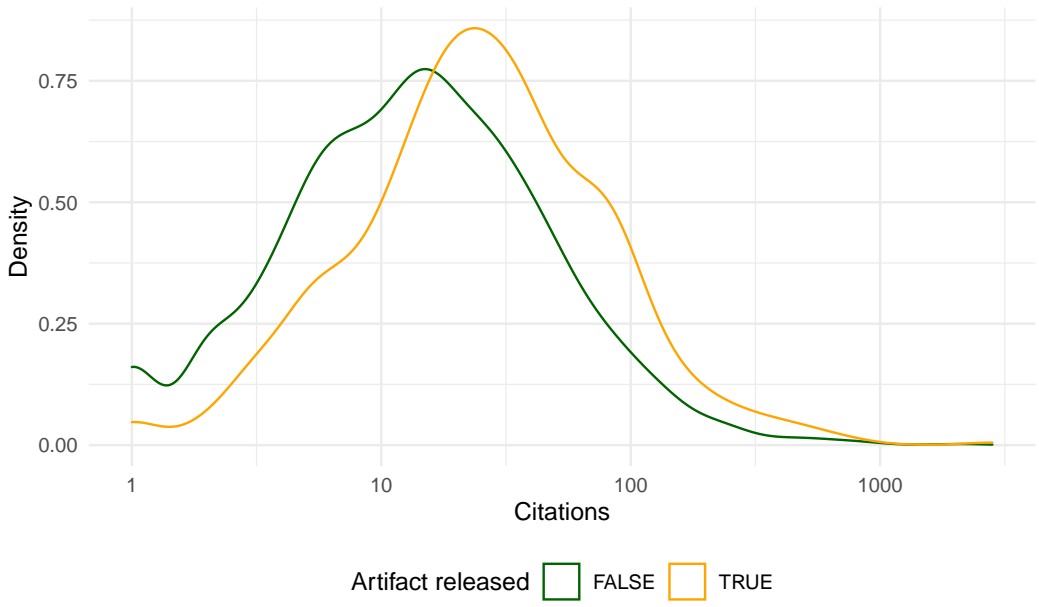

**Figure 3** Density plot of paper citations 42 months after publication (log-scale).

**Table 3** Median citations by class of artifact URLs for extant artifacts.

| Location | Count | Median citations |
| --- | --- | --- |
| Repository | 477 | 25 |
| Academic | 81 | 24 |
| Other | 67 | 27 |
| ACM | 44 | 15 |
| Filesharing | 6 | 35 |

appear to contradict the premise that such badges are associated with increased artifact sharing, as has been found in other fields (*Baker, 2016a*).

Finally, we can also break down the citations per paper grouped by the type of location for the artifact and by its organization, examining medians because of the outsize effects of outliers (Table 3). The three major location categories do not show significant differences in citations, and the last two categories may be too small to ascribe statistical significance to their differences.

## Accessibility

One commonly used set of principles to assess research software artifacts is termed FAIR: findability, accessibility, interoperability, and reusability (*Hong et al., 2021*; *Wilkinson et al., 2016*). We have overviewed the findability aspect of artifacts in the statistics of how many of these were linked or found *via* a Web search. The reusability and interoperability of artifacts unfortunately cannot be assessed with the current data. But we can address some of our secondary research questions by analyzing the accessibility of artifacts in depth.

As mentioned previously, 13.3% of released artifacts are already inaccessible, a mere $\approx 3.5$ years after publication. Most of the artifacts in our dataset were published in code repositories, predominantly github, that do not guarantee persistent access or even universal access protocols such as digital object identifiers (DOI). However, only 2.3% of the "Repository" artifacts were inaccessible. In contrast, 22.2% of the artifacts in university pages have already expired, likely because they had been hosted by students or faculty that have since moved elsewhere. Also, a full half of the artifacts on file-sharing sites such as Dropbox or Google Drive are no longer there, possibly because these are paid services or free to a limited capacity, and can get expensive to maintain over time.

Accessibility is also closely related to the findability of the artifact, which in the absence of artifact DOIs in our dataset, we estimate by looking at the number of papers that explicitly link to their artifacts. The missing (expired) artifacts consisted of a full 31.1% of the papers with no artifact link, compared to only 8.7% for papers that linked to them ($\chi^2 = 49.15$, $p < 10^{-9}$).

Another related question to artifact accessibility is how accessible is the actual paper that introduced the artifact, which may itself be associated with higher citations (*Gargouri et al., 2010*; *McCabe & Snyder, 2015*; *McKiernan et al., 2016*; *Tahamtan, Afshar & Ahamdzadeh, 2016*). A substantial proportion of the papers (23.1%) were published in 15 open-access conferences. Other papers have also been released openly as preprints or *via* other means. One way to gauge the availability of the paper's text is to look it up on GS and see if an accessible version (eprint) is linked, which was recorded in our dataset. Of the 2,439 papers, 91.8% displayed at some point an accessible link to the full text on GS. Specifically, of the papers that released artifacts, 96.7% were associated with an eprint as well, compared to 90% of the papers with no artifacts ($\chi^2 = 29$, $p < 10^{-7}$).

Moreover, our dataset includes not only the availability of an eprint link on GS, but also the approximate duration since publication (in months) that it took GS to display this link, offering a quantitative measure of accessibility speed. It shows that for papers with artifacts, GS averaged approximately 4 months post-publication to display a link to an eprint, compared to 5.8 months for papers with no artifacts ($t = -5.48$, $p < 10^{-7}$). Both of these qualitative and quantitative differences are statistically significant, but keep in mind that the accessibility of papers and artifacts are not independent: some conferences that encouraged artifacts were also open-access, particularly those with the ACM. Another dependent covariate with accessibility is citations; several studies suggested that accessible papers are better cited (*Bernius & Hanauske, 2009*; *Niyazov et al., 2016*; *Snijder, 2016*), although others disagree (*Calver & Bradley, 2010*; *Davis & Walters, 2011*; *McCabe & Snyder, 2015*). This dependence may explain part of the higher citability of papers with artifacts, as elaborated next.

## Covariate analysis

Having addressed the relationships between artifacts and citations, we can now explore relationships between additional variables from this expansive dataset.

### Awards

Many conferences present competitive awards, such as "best paper", "best student paper", "community award", *etc.* Of the 2,439 total papers, 4.7% received at least one such award. Papers with artifacts are disproportionately represented in this exclusive subset (39.5% *vs.* 27.1% in non-award papers; $\chi^2 = 7.71$, $p < 0.01$).

Again, it is unclear whether this relationship is causal since the two covariates are not entirely independent. For example, a handful of awards specifically evaluated the contribution of the paper's artifact. Even if the relationship is indeed causal, its direction is also unclear, since 20% of award papers with artifacts did not link to it in the paper. It is possible that these papers released their artifacts after winning the award or because of it.

### Textual properties

Some of the textual properties of papers can be estimated from their full text using simple command-line tools. Our dataset includes three such properties: the length of each paper in words, the number of references it cites, and the existence of a system's moniker in the paper's title.

The approximate paper length in words and the number of references turn out to be positively associated with the release of an artifact. Papers with artifacts average **more pages** than papers without (13.98 *vs.* 12.4; $t = 8.24$, $p < 10^{-9}$), **more words** (11757.36 *vs.* 10525.22; $t = 7.86$, $p < 10^{-9}$), and **more references** (32.31 *vs.* 28.71; $t = 5.25$, $p < 10^{-6}$). Keep in mind, however, that longer papers also correspond to more references ($r = 0.48$, $p < 10^{-9}$), and are further confounded with specific conference factors such as page limits.

As previously mentioned, many systems papers introduce a new computer system, often as software. Sometimes, these papers name their system by a moniker, and their title starts with the moniker, followed by a colon and a short description (*e.g.*, "Widget: An Even Faster Key-Value Store"). This feature is easy to extract automatically for all paper titles.

We could hypothesize that a paper that introduces a new system, especially a named system, would be more likely to include an artifact with the code for this system, quite likely with the same repository name. Our data support this hypothesis. The ratio of artifacts released in papers with a labeled title, 41.9%, is nearly double that of papers without a labeled title, 22.8% ($\chi^2 = 84.23$, $p < 10^{-9}$).

The difficulty to ascribe any causality to these textual relationships could mean that there is little insight to be gained from them. But they can clue the paper's reader to the possibility of an artifact, even if one is not linked in the paper. Indeed, they accelerated the manual search for such unlinked artifacts during the curation of the data assisted for this study.

### Conference prestige

Next, we look at conference-specific covariates that could represent how well-known or competitive a conference is. In addition to textual conference factors, these conference metrics may also be associated with higher rates of artifact release.

Several proxy metrics for prestige appear to support this hypothesis. Papers with released artifacts tend to appear in conferences that average a **lower acceptance rate** (0.21 *vs.* 0.24; $t = -6.28$, $p < 10^{-9}$), **more paper submissions** (360.5 *vs.* 292.45; $t = 6.33$, $p < 10^{-9}$),

**higher historical mean citations per paper** (16.6 *vs.* 14.96; $t = 3.09$, $p < 0.01$), and a **higher h5-index** from GS metrics (46.04 *vs.* 41.04; $t = 6.07$, $p < 10^{-8}$). Also note that papers in conferences that offered some option for author response to peer review (often in the form of a rebuttal) were slightly more likely to include artifacts, perhaps as a response to peer review ($\chi^2 = 2.03$, $p = 0.15$).

To explain these relationships, we might hypothesize that a higher rate of artifact submission would be associated with more reputable conferences, either because artifact presence contributes to prestige, or because more rigorous conferences are also more likely to expect such artifacts. Observe, however, that some of the conferences that encourage or require artifacts are not as competitive as the others. For example, OOPSLA, with the highest artifact rate, had an acceptance rate of 0.3, and SLE, with the fourth-highest artifact rate, had an acceptance rate of 0.42. The implication here is that it may not suffice for a conference to actively encourage artifacts for it to be competitive, but a conference that already is competitive may also attract more artifacts.

### Regression model

Finally, we combine all of these factors to revisit in depth our primary research interest: the effect of artifact sharing on citations. We already observed a strong statistical association between artifact release and higher eventual citations. As cautioned throughout this study, such associations are insufficient to draw causal conclusions, primarily because there are many confounding variables, most of which relating to the publishing conference. These confounding factors could provide a partial or complete statistical explanation to differences in citations beyond artifact availability.

In other words, papers published in the same conference might exhibit strong correlations that interact or interfere with our response variable. One such factor affecting paper citations is time since publication, which we control for by measuring all citations at exactly the same interval, 42 months since the conference's official start. Another crucial factor is the field of study—which we control for by focusing on a single field—while providing a wide cross-section of the field to limit the effect of statistical variability.

There are also numerous less-obvious paper-related factors that have shown positive association with citations, such as review-type studies, fewer equations, more references, statistically significant *positive* results, papers' length, number of figures and images, and even more obscure features such as the presence of punctuation marks in the title. We can attempt to control for such confounding variables when evaluating associations by using a multilevel model. To this end, we fit a linear regression model of citations as a function of artifact availability, and then add predictor variables as controls, observing their effect on the main predictor. The response variable we model for is *ln(citations)* instead of citations, because of the long tail of their distribution. We also omit the 49 papers with zero citations to improve the linear fit with the predictors.

In the baseline form, fitting a linear model of the log-transformed citations as a function of only artifact released yields an intercept (baseline log citations) of 2.6 and a slope of 0.59, meaning that releasing an artifact adds approximately 81% more citations to the paper, after exponentiation. The *p*-value for this predictor is exceedingly low (less than $2 \times 10^{-16}$) but

[2]Papers with no eprint available at the time of this writing were assigned an arbitrary time to eprint of 1,000 months, but the regression analysis was not particularly sensitive to this choice.

the simplistic model only explains 4.61% of the variance in citations (Adjusted $R^2 = 0.046$). The Bayesian Information Criterion (BIC) for this model is 7693.252, with 2388 degrees of freedom (df).

We can now add various paper covariates to the linear model in an attempt to get more precise estimates for the artifact released predictor, by iteratively experimenting with different predictor combinations to minimize BIC using stepwise model selection (*García-Portugués, 2021*, Ch. 3). The per-paper factors considered were: **paper length** (words), **number of coauthors**, **number of references**, **colon in the title**, **award** given, and **accessibility speed** (months to eprint[2]).

It turns out that all these paper-level factors except award given have a statistically significant effect on citations, which brings the model to an increased adjusted $R^2$ value of 0.285 and a BIC of 7028.07 ($df = 2,380$). However, the coefficient for artifact released went down to 0.35 (42% relative citation increase) with an associated *p*-value of $3.8 \times 10^{-13}$.

Similar to paper variables, some author-related factors such as their academic reputation, country of residence, and gender have been associated with citation count (*Tahamtan, Afshar & Ahamdzadeh, 2016*). We next enhance our linear model with the following predictor variables (omitting 451 papers with NA values):

- Whether all the coauthors with a known affiliation came from the same country (*Puuska, Muhonen & Leino, 2014*).
- Is the lead author affiliated with the United States (*Gargouri et al., 2010*; *Peng & Zhu, 2012*)?
- Whether any of the coauthors was affiliated with one of the top 50 universities per www.topuniversities.com (27% of papers) or a top company (if any author was affiliated with either (Google, Microsoft, Yahoo!, or Facebook: 18% of papers), based on the definitions of a similar study (*Tomkins, Zhang & Heavlin, 2017*).
- Whether all the coauthors with a known affiliation came from industry.
- The gender of the first author (*Frachtenberg & Kaner, 2021*).
- The sum of the total past publications of all coauthors of the paper (*Bjarnason & Sigfusdottir, 2002*).
- The maximum h-index of all coauthors (*Hurley, Ogier & Torvik, 2014*).

[3]Note that the past publication counts and h-index are correlated ($r = 0.6$), ($p < 10^{-9}$), so one may cancel the other out.

Only the maximum h-index and top-university affiliation had statistically significant coefficients, but hardly affected the overall model.[3] These minimal changes may not justify the increased complexity and reduced data size of the new model (because of missing data), so for the remainder of the analysis, we ignore author-related factors and proceed with the previous model.

We can now add the last level: venue factors. Conference (or journal) factors—such as the conference's own prestige and competitiveness—can have a large effect on citations, as discussed in the previous section. Although we can approximate some of these factors with some metrics in the dataset, there may also be other unknown or qualitative conference factors that we cannot model. Instead, to account for conference factors we next build a mixed-effects model, where all the previously mentioned factors become fixed effects and the conference becomes a random effect (*Roback & Legler, 2021*, Ch. 8).

**Table 4   Estimated parameters for final multilevel mixed-effects model of ln(citations).**

| Factor | Coefficient | *p*-value |
|---|---|---|
| Intercept | 1.74166 | 6.6e−31 |
| Artifact released | 0.29357 | 3.6e−10 |
| Award given | 0.00002 | 4.7e−02 |
| Months to eprint | −0.00048 | 8.8e−09 |
| References number | 0.00907 | 1.3e−08 |
| Coauthors number | 0.06989 | 1.6e−19 |
| Colon in title | 0.13369 | 1.4e−03 |

This last model does indeed reduce the relative effect of artifact release on citations to a coefficient of 0.29 (95% confidence interval: 0.2–0.39). But this coefficient still represents a relative citation increase of about a third for papers with released artifacts (34%), which is substantial. We can approximate a *p*-value for this coefficient *via* Satterthwaite's degrees of freedom method using R's `lmerTest` package (*Kuznetsova, Brockhoff & Christensen, 2017*), which is also statistically significant at $3.5825 \times 10^{-10}$. The parameters for this final model are enumerated in Table 4. The only difference in paper-level factors is that award availability has replaced word count as a significant predictor, but realistically, both have a negligible effect on citations.

# DISCUSSION

## Implications

The regression model described in the preceding section showed that even with multiple controlling variables we observe a strong association between artifact release and citations. We can therefore ask, does this association allow for any causal or practical inferences? This association may still not suffice to claim causation due to hidden variables (*Lewis, 2018*), but it does support the hypothesis that releasing artifacts can indeed improve the prospects of a systems research paper to achieve wider acceptance, recognition, and scientific impact.

One implication of this model is that even if we assume no causal relation between artifact sharing and higher citation counts, the association is strong enough to justify a change in future scientometric studies of citations. Such studies often attempt to control for various confounders when attempting to explain or predict citations, and this strong link suggests that at least for experimental and data-driven sciences, the sharing of research artifacts should be included as an explanatory variable.

That said, there may be a case for a causal explanation with a clear direction after all. First, the model controls for many of the confounding variables identified in the literature, so the possibility of hidden, explanatory variables is diminished. Second, there is a clear temporal relationship between artifact sharing and citations. Artifact sharing invariably accompanies the publication of a paper, while its citations invariable follow months or years afterward. It is therefore plausible to expect that citation counts are influenced by artifact sharing and not the other way around.

If we do indeed assume causality between the two, then an important, practical implication also arises from this model, especially for authors wishing to increase their work's citations. There are numerous factors that authors cannot easily control, such as their own demographic factors, but fortunately, these turn out to have insignificant effects on citations. Even authors' choice of a venue to publish in, which does influence citations, can be constrained by paper length, scope match, dates and travel, and most importantly, the peer-review process that is completely outside of their control. But among the citation factors that authors can control, the most influential one turns out to be the sharing of research artifacts.

A causal link would then provide a simple lever for systems authors to improve their citations by an average of some 34%: share and link any available research artifacts. Presumably, authors attempting to maximize impact already work hard to achieve a careful study design, elaborate engineering effort, a well-written paper, and acceptance at a competitive conference. The additional effort of planning for and releasing their research artifact should be a relatively minor incremental effort that could improve their average citation count. If we additionally assume causality in the link between higher artifact sharing rates and acceptance to more competitive conferences, the effect on citations can be compounded.

Other potential implications of our findings mostly agree with our intuition and with previous findings in related studies, as described in the related-work. For example, all other things being equal, papers with open access and with long-lasting artifacts receive more citations.

Two factors that do not appear to have a positive impact on citations, at least in our dataset, are the receipt of artifact badges or the linking of artifacts in the paper. This is unfortunate because it implicitly discourages standardized or searchable metadata on artifacts, which is critical for studies on their effect, as described next.

## Threats to validity

Perhaps the greatest challenge in performing this study or in replicating it is the fact that good metadata on research artifacts is either nonexistent or nonstandard. There is currently no automated or even manual methodology to reliably discover which papers shared artifacts, how they were they shared, and how long did they survive. There are currently several efforts underway to try to standardize artifact metadata and citation, but for this current study, the validity and scalability of the analysis hinge on the quality of the manual process of data collection.

One way to address potential human errors in data collection and tagging is to collect a sizeable dataset—as was attempted in this dataset—so that such errors disappear in the statistical noise. Although a large-enough number of artifacts was identified for statistical analysis, there likely remain untagged papers in the dataset that did actually release an artifact (false negatives). Nevertheless, there is no evidence to suggest that their number is large or that their distribution is skewed in some way as to bias statistical analyses. Moreover, since the complete dataset is (naturally) released as an artifact of this paper, it can be enhanced and corrected over time.

Additionally, there is the possibility of errors in the manual process of selecting conferences, importing data about papers and authors, disambiguating author names, and identifying the correct citation data on GS. In the data-collection process, we have been careful to cross-validate the data we input against the one found in the official proceedings of each conference, as well as the data that GS recorded, and reconciled any differences we found.

Citation metrics were collected from the GS database because it includes many metrics and allows for manual verification of the identity of each author by linking to their homepage. This database is not without its limitations, however. It does not always disambiguate author names correctly, and it tends to overcount publications and citations (*Halevi, Moed & Bar-Ilan, 2017*; *Harzing & Alakangas, 2016*; *Martin-Martin et al., 2018*; *Sugimoto & Lariviere, 2018*). The name disambiguation challenge was addressed by manually verifying the GS profiles of all researchers and ensuring that they include the papers from our dataset. Ambiguous profiles were omitted from our dataset. As for citation over-counting, note that the absolute number of citations is immaterial to this analysis, only the difference between papers with and without artifacts. Assuming GS overcounts both classes of papers in the same way, it should not materially change the conclusions we reached.

Our dataset also does not include data specific to self-citations. Although it is possible that papers with released artifacts have different self-citations characteristics, thus confounding the total citation count, there is no evidence to suggest such a difference. This possibility certainly opens up an interesting question for future research, using a citation database with reliable self-citation information (unlike GS).

## RELATED WORK

This paper investigates the relationship between research artifacts and citations in the computer systems field. This relationship has been receiving increasingly more attention in recent years for CS papers in general. For example, a new study on software artifacts in CS research observed that while artifact sharing rate is increasing, the bidirectional links between artifacts and papers do not always exist or last very long, as we have also found (*Hata et al., 2021*). Some of the reasons that researchers struggle to reproduce experimental results and reuse research code from scientific papers are the continuously changing software and hardware, lack of common APIs, stochastic behavior of computer systems, and a lack of a common experimental methodology (*Fursin, 2021*), as well as copyright restrictions (*Stodden, 2008*).

Software artifacts have often been discussed in the context of their benefits for open, reusable, and reproducible science (*Hasselbring et al., 2019*). Such results have led more CS organizations and conferences to increase adoption of artifact sharing and evaluation, including a few of the conferences evaluated in this paper (*Baker, 2016b*; *Dahlgren, 2019*; *Hermann, Winter & Siegmund, 2020*; *Saucez, Iannone & Bonaventure, 2019*). One recent study examined specifically the benefit of software artifacts for higher citation counts (*Heumüller et al., 2020*). Another study looked at artifact evaluation for CS papers and

found a small but positive correlation with higher citations counts for papers between 2013 and 2016 (*Childers & Chrysanthis, 2017*).

When analyzing the relationship between artifact sharing and citations, one must be careful to consider the myriad possibilities for confounding factors, as we have in our mixed-effects model. Many such factors have been found to be associated with higher citation counts. Some examples relating to the author demographics include the authors' gender (*Frachtenberg & Kaner, 2021*; *Tahamtan, Afshar & Ahamdzadeh, 2016*), country of residence (*Gargouri et al., 2010*; *Peng & Zhu, 2012*; *Puuska, Muhonen & Leino, 2014*), affiliation (*Tomkins, Zhang & Heavlin, 2017*), and academic reputation metrics (*Hurley, Ogier & Torvik, 2014*; *Bjarnason & Sigfusdottir, 2002*). Other factors were associated with the publishing journal or conference, such as the relative quality of the article and the venue (*McCabe & Snyder, 2015*) and others still related to the papers themselves, such as characteristics of the titles and abstracts, characteristics of references, and length of paper (*Tahamtan, Afshar & Ahamdzadeh, 2016*).

Among the many paper-related factors studied in relation to citations is the paper's text availability, which our data shows to be also linked with artifact availability. there exists a rich literature examining the association between a paper's own accessibility and higher citation counts, the so-called "OA advantage" (*Bernius & Hanauske, 2009*; *Davis & Walters, 2011*; *Sotudeh, Ghasempour & Yaghtin, 2015*; *Wagner, 2010*).

For example, Gargouri et al. found that articles whose authors have supplemented subscription-based access to the publisher's version with a freely accessible self-archived version are cited significantly more than articles in the same journal and year that have not been made open (*Gargouri et al., 2010*). A few other more recent studies and reviews not only corroborated the OA advantage but also found that the proportion of OA research is increasing rapidly (*Breugelmans et al., 2018*; *Fu & Hughey, 2019*; *McKiernan et al., 2016*; *Tahamtan, Afshar & Ahamdzadeh, 2016*). The actual amount by which open access improves citations is unclear, but one recent study found the number to be approximately 18% (*Piwowar et al., 2018*), which means that higher paper accessibility on its own is not enough to explain all of the citation advantage we identified for papers with available artifacts.

Turning our attention specifically to the field of systems, we might expect that many software-based experiments should be both unimpeded and imperative to share and reproduce (*Ince, Hatton & Graham-Cumming, 2012*). But instead we find that many artifacts are not readily available or buildable (*Collberg & Proebsting, 2016*; *Freire, Bonnet & Shasha, 2012*; *Heumüller et al., 2020*; *Krishnamurthi & Vitek, 2015*). A few observational studies looked at artifact sharing rates in specific subfields of systems, such as software engineering (*Childers & Chrysanthis, 2017*; *Heumüller et al., 2020*; *Timperley et al., 2021*) and computer architecture (*Fursin & Lokhmotov, 2011*), but none that we are aware of have looked across the entire field.

Without directly comparable information on artifact availability rates in all of systems or in other fields, it is impossible to tell whether the overall rate of papers with artifacts in our dataset, 27.7%, is high or low. However, within the six conferences that evaluated artifacts, 42.86% of papers released an artifact, a very similar rate to the $\approx 40\%$ rate found

in a study of of a smaller subset of systems conferences with an artifact evaluation process (*Childers & Chrysanthis, 2017*).

In general, skimming the papers in our dataset revealed that many "systems" papers do in fact describe the implementation of a new computer system, mostly in software. It is plausible that the abundance of software systems in these papers and the relative ease of releasing them as software artifacts contributes directly to this sharing rate, in addition to conference-level factors.

## CONCLUSION

Several studies across disparate fields found a positive association between the sharing of research artifacts and increased citation of the research work. In this cross-sectional study of computer systems research, we also observed a strong statistical relationship between the two, although there are numerous potential confounding and explanatory variables to increased citations. Still, even when controlling for various paper-related and conference-related factors, we observe that papers with shared artifacts receive approximately one-third more citations than papers without.

Citation metrics are a controversial measure of a work's quality, impact, and importance, and perhaps should not represent the sole or primary motivation for authors to share their artifacts. Instead, authors and readers may want to focus on the clear and important benefits to science in general, and to the increased reproducibility and credibility of their work in particular. If increased citation counts are not enough to incent more systems authors to share their artifacts, perhaps conference organizers can leverage their substantial influence to motivate authors. Although artifact evaluation can represent a nontrivial additional burden on the program committee, our data show that it does promote higher rates of artifact sharing.

While many obstacles to the universal sharing of artifacts still remain, the field of computer systems does have the advantage that many—-if not most—of its artifacts come in the form of software, which is easier to share than artifacts in other experimental fields. It is therefore not surprising that we find the majority of shared and extant artifacts in computer systems hosted on github.com, a highly accessible source-code sharing platform. That said, a high artifact sharing rate is not enough for the goals of reproducible science, since many of the shared artifacts in our dataset have since expired or have been difficult to locate.

Our analysis found that both the findability and accessibility of systems artifacts can decay significantly even after only a few years, especially when said artifacts are not hosted on dedicated open and free repositories. Conference organizers could likely improve both aspects by requiring—and perhaps offering—standardized tools, techniques, and repositories, in addition to the sharing itself. The ACM has taken significant steps in this direction by not only standardizing various artifact badges but also offering its own supplementary material repository in its digital library. A few conferences in our dataset, like SC, are taking another step in this direction by also requesting a standardized artifact description appendix and review for every technical paper, including a citeable link to the research artifacts.

To evaluate the impact of such efforts, we must look beyond the findability and accessibility of artifacts, as was done in this study. In future work, this analysis can be expanded to the two remaining aspects of the FAIR principles: interoperability and reusability, possibly by incorporating input from the artifact review process itself. The hope is that as the importance and awareness of research artifacts grows in computer systems research, many more conferences will require and collect this information, facilitating not only better, reproducible research, but also a better understanding of the nuanced effects of software artifact sharing.

## ACKNOWLEDGEMENTS

I wish to thank Prof. Kelly McConville of Reed College for her thoughtful and patient assistance with the statistical analysis.

### Funding
The author received no funding for this work.

### Competing Interests
The authors declare there are no competing interests.

### Author Contributions
- Eitan Frachtenberg conceived and designed the experiments, performed the experiments, analyzed the data, performed the computation work, prepared figures and/or tables, authored or reviewed drafts of the paper, and approved the final draft.

### Data Availability
All source and data are available in the Supplemental Files.

### Supplemental Information
Supplemental information for this article can be found online at http://dx.doi.org/10.7717/peerj-cs.887#supplemental-information.

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
