# Peer review of "Research artifacts and citations in computer systems papers"

_PeerJ Computer Science, doi:10.7717/peerj-cs.887_

## Round 0.1 · original submission · Major Revisions

Overall, all reviewers agree that the research topic is relevant. However, they think that there are several points to improve in your article: (i) Summarize the main findings of the study in the Introduction, (ii) Clearly define the research goals/ hypotheses, (iii) Create a Related Work section, (iv) State the study population and sample investigated, (v) Define the protocol of the study, (vi) Promote the Data/Software Availability with their own DOI at Zenodo or OSF. In summary, the author should clearly define the design of his study to support his findings. Please consider these comments in your new version.

Reviewer 1 ·

Basic reporting

1.1. Clear and unambiguous, professional English used throughout.
I find the English acceptable for publication. However, I suggest adding some definitions and unifying some concepts. I found it difficult to understand whether some expressions were used interchangeably or not, which turned out to make it difficult to realize the scope of the paper regarding the type of software included in the study. In particular,
* Is “computer systems” used as a synonym of “computer science”, “software”, “system software”, some of them or none?
* Does system software correspond to software used as a platform to develop other software including operating systems, software languages and/or utility software?
* Is the paper about system software in general or only system software created as part of a research process? Is the paper about software artifacts or research artifacts (which also include, for instance, data)?
* What is the field the author refers to? Is it computer science, computer systems, system software, research software?

Some typos:
* Line 11, “research’s results”, possibly no need of the possessive form
* Line 60, “are valued an are an important contribution”, possibly “and” rather than “an”
* Line 443 and 444, “many—-f not most—”, possible “if” rather than “f”
I suggest a full proof-reading to catch and fix those issues.

1.2. Literature references, sufficient field background/context provided.
The article is accompanied with a good number of references. However, the author should double check the citation information. For instance
* Line 472, reference ACM, URL not working (https://www.acm.org/publications/policies/artifact-review473 and-badging-current).
* Whenever possible, even for web pages, please include information about the author and published date.
* The reference used for FAIR applied to research software (line 319, Katz et al., 2021) corresponds to a (partial) outcome of a Research Data Alliance Working Group and could be updated to the now first formal output produced by this group (see DOI:10.15497/RDA00065 and corresponding PDF https://www.rd-alliance.org/system/files/FAIR4RS_Principles_v0.3_RDA-RFC.pdf)
* I suggest also including a reference to the code supporting this publication. I know it is provided as supplementary material but getting a permanent identifier together with citation data for this particular release is also possible (and slowly becoming a common practice). I would also like to see the GitHub pages working and linked to from the publication (possibly with a note regarding differences that might appear as the repo evolves post-publication).

1.3. Professional article structure, figures, tables. Raw data shared.
The article structure is appropriate as well as figures and tables.
Raw (and processed) data is included in the supplementary materials.

1.4. Self-contained with relevant results to hypotheses.
It was possible to read the article on its own.
Results are aligned to hypotheses.

1.5. Formal results should include clear definitions of all terms and theorems, and detailed proofs.
Code to process raw data and analyze results is shared, including a set of instructions on how to use it. The presentation of results is clear, detailed and supported by the presented data.

Experimental design

2.1. Original primary research within Aims and Scope of the journal.
The article aligns to the aims and scope of the journal.

2.2. Research question well defined, relevant & meaningful. It is stated how research fills an identified knowledge gap.
Yes

2.3. Rigorous investigation performed to a high technical & ethical standard.
It looks like.

2.4. Methods described with sufficient detail & information to replicate.
It looks like.

Validity of the findings

3.1. Impact and novelty not assessed. Meaningful replication encouraged where rationale & benefit to literature is clearly stated.
Replication possible thanks to the available code and basic set of instructions on how to run it

3.2. All underlying data have been provided; they are robust, statistically sound, & controlled.
Raw data has been provided.

3.3. Conclusions are well stated, linked to original research question & limited to supporting results.
I would say yes.

Additional comments

I find this article of broad interest for the research community. Sharing software and in general research artifacts is slowly becoming a common practice aligned to Open Science and the FAIR principles. I hope to see more research on this topic.

As the authors stated, one of the main challenges was the curation of the data which was mainly manual. Although out of the scope of this paper, I hope research in this area will motivate not only conferences but also preprints and journals to be more proactive regarding the metadata accompanying research artifacts.

Reviewer 2 ·

Basic reporting

[Introduction]
The list of detailed research questions presented in the Introduction contrasts with the regular scope of an initial section. These questions should be properly justified and connected, which should be done in a further section. Alternatively, I suggest the authors expanding the "interesting" question (line 66) by exemplifying other possible indicators that would be influenced by artifact sharing.

I suggest the authors summarize the main findings of the study in the Introduction.

[Background and Related Work]
The author cites several references addressing open science and initiatives promoting artifact sharing, which address the research background. However, I missed a section discussing related work on the topic. For instance, previous investigations/discussions on the maturity of open science in the field.

[Figures/Tables]
I suggest presenting the Figure 1 data using tables or grouping conferences by frequency. Yet regarding Figure 1, it is not clear to me the relevance of identifying the "Organization" of the conferences. For instance, several conferences continuously alternate between ACM and IEEE.

I understand Table 1 should adopt a better-contextualized criterion than alphabetical order for listing conferences. For instance, the author may use the number of papers or the acceptance rate.

Each conference has a variable time of running/publication, which may even lead to its publishing in the next year. I would like to understand how the author reached "42 months of publication" for all publications and why he opted by following this gold number.

[Verifiability]
I was expecting a user-friendly dataset available from a research paper addressing the artifacts' availability in research papers.

Experimental design

Before reading about the study results (Section 3), I was expecting a Section reporting the study design. After reading the entire section of "Data and Methods" (Section 2), I could not find any reference to the research goals/ hypotheses. Besides, the author should properly address the study population and sample investigated.

The text of section 2 is too monolithic and hard to follow. It does not address the basic content expected from a study design section. After reading it, I only got a vague idea about what the authors had done (execution) and the main data types they collected.

It is not clear the criteria followed by the author to reach the "hand-curated collection" of 56 peer-reviewed "systems conferences." Even if the sample was established by convenience from google scholar, the authors should let it clear in the paper.

I could not reach what does the author mean by "systems conferences." Besides, I am afraid about the lack of representativeness of the addressed in the study. For instance, I could found only one relevant conference from the Software Engineering field.

It is not clear the methodology followed for identifying artifacts in the papers. Besides, I understand that artifacts available in research papers should also address their quality. I recommend the authors planning and performing an evaluation in this direction.

Validity of the findings

Section 3.2 illustrates my difficulty in understanding the research methodology. It is a subsection about results. However, it starts by presenting a new research question: "does the open availability of an artifact affect the citations of a paper in computer systems?". Besides, this RQ is presented as "the main research question of this paper." Contrastingly, several of the eight research questions presented in the paper Introduction are barely (or clearly) addressed in the paper.

The findings of the paper are not discussed. For instance, what are the possible implications of these findings for the practice of sharing artifacts? Which are possible ways for filling the gaps observed? The section 4.1 "Discussion" basically summarize the several statistical tests and analysis performed (4.1). However, even here I could not see a clear connection among them. Besides, without a proper research plan, I could not get why all these tests and analyses were performed.

Only at the end of Section 5, I could observe some links between the study results and some implications.

Additional comments

Unfortunately, the lack of properly reporting basic content expected from a research paper hampered me from performing a more accurate review and indicating further suggestions for improvement. Despite that, I recognize the relevance of the research topic. Therefore, I encourage the author to keep on with this research. Maybe the author would rethink the design of this study as a systematic review addressing other research artifacts beyond the papers.

·

Basic reporting

This article is well-written and addresses statistically the very important question if sharing of software artifacts has an effect on citation metrics of computer science, as well as the need for proper archiving of software artefacts for longevity.

The argument is well put forward and supported by a thorough data investigation for the given sample set.

A couple of smaller typos were found

As the article points out, the availability of software artefacts is important - however this article itself does not have an explicit "Data and software availability" section - which would be very valuable given the data quality and reproducibility provided for this manuscript.

I would also expect the data to be archived in Zenodo with a DOI rather than be a fluctual GitHub reference - I know the author have also uploaded a snapshot of said GitHub repository along with the article - but this would not have been detected by their own measures.

As the data seems to be used for multiple articles under review, a separate version-less Zenodo DOI would be able to bring these articles together for describing different aspects of the same data (e.g. using Related Identifiers mechanism in Zenodo).Only one of these other articles are cited from this manuscript.

Kelly McConville is acknowledged for assisting with the statistical analysis - this article is quite strong on statistics - I might have added McConville as co-author to recognize this fact.

Experimental design

This research is rigourous and the experimental design is well explained.

Validity of the findings

I have not assessed the choices of statistical methods, but the authors have been thorough throughout, e.g. providing p-values and describing method choices. The actual values and figures are also calculated dynamically by the R markdown for the manuscript, and so can be reproducibly verified.

In order to verify the reproducibility I had some challenges in getting the R environment set up - however the author quickly responded to my GitHub issue and they added detail installation instructions which helped me.

The findings are based on a large subste of the CS System conferences - perhaps more could be added on why these particular conferences (e.g. practicallity of access, notability, familiarity to the authors). The authors have done a good selection time-wise (all considered data in 2017) in order to consider citations building over time.

The data set can be hard to navigate and use as it is shared across multiple publications and corresponding R markdown environments. README files have been provided, but the manuscript itself does not detail which of the data is relevant to this particular manuscript. For instance it is possible for someone proficient in R to analyse the R Markdown of pubs/artifact/artifact.Rmd where the statistical analysis is embedded.

The data/ CSV files are also well documented in that they each have a little file describing their schema - some of these state their provenance, which is nice.

Additional comments

Hi, I am Stian Soiland-Reyes http://orcid.org/0000-0001-9842-9718 and believe in open reviews.

This article is a very welcome addition to the CS field, where sharing of software artefacts have only recently emerged as a practice, catcing up on fields like bioinformatics.

The key message from this article for me is showing evidence for CS authors that sharing artefacts is beneficial for science and for their article's to be cited.

I therefore think this is a very important article for PeerJ CS to publish.

My recommendation would have been Accept except for that the article should more clearly promote its Data/Software Availability as well as archiving those with their own DOI at Zenodo or similar repository.

---

## Round 0.2 · Major Revisions

The authors made several improvements in the article. The readability was improved and many decisions are now well explained. However, one of the reviewers still believes that some improvements are needed: (i) interpretation of the implications of findings, (ii) threats to validity discussion, and (iii) explanation about the research questions. Please consider these topics when preparing the new version of your article.

Reviewer 2 ·

Basic reporting

I appreciate most of the improvements made by the author to the paper. Now, the paper reading flows better. The main research goal and study hypothesis are clear. Some design decisions are now better contextualized. ​However, there are still relevant remaining issues to be addressed. In the following, I summarize my key recommendations for this paper.

1- The research steps should be completely described, especially in the case of following a systematic design (which is not still clear to me).
2- The author should contextualize the detailed research questions and corresponding data analysis procedures, considering main study goals and hypotheses.
3- Discussions should focus on interpreting more the implications of the study findings rather than adding more ad hoc data analysis and statistical tests.
4- The author should significantly improve the discussions about threats to validity.

Experimental design

The author does not report systematic steps followed for gathering papers' data. For instance, "....several search terms were used to assist in identifying artifacts, such as “github,” “gitlab,” “bitbucket,”, “sourceforge,” and “zenodo” for repositories; variants of “available,” “open source,” and “download” for links; and variations of “artifact,” “reproducibility,” and “will release” for indirect references." Besides, it is not clear how the author checked out whether a paper received an award.

At the beginning of the paper, several research questions are presented without a proper justification. These detailed RQ should be addressed and contextualized in the proper section.

I understand that the authors followed a partially ad-hoc process for gathering papers' data without double-checking, instead of following a systematic one. If true, the author should explicit these relevant threats to validity. Otherwise, the author should report step-by-step how they reached each data to led the package reproducible.

Validity of the findings

I could not find any mention to double-checking on the gathered data. It is relevant at least fo support data gathering through non-systematic steps.

What was the normality test applied? p-value? Yet about normality, the authros state that the data was normalized. However, after that the author argue that "...omit the 49 papers with zero citations to improve the linear fit with the predictors." Outliers? And what about those with >2000 citations?

"...It is worth noting, however, that most of the source-code repositories in these
artifacts showed no development activity—commits, forks, or issues—after the publication of their paper, suggesting limited impact for the artifacts alone". I disagree about this example of poor impact. Please note that commits and forks are typically made for source code, which is not necessarily the case of research data.

I miss a discussion of the findings towards causality (which seems to be unfeasible) and more statistical tests (that does not help to understand the original results. Some of the complementary analysis seems to address irrelevant papers' features (at least without proper contextualization). For instance: "Incidentally, papers with released artifacts also tend to incorporate significantly more references themselves (mean: 32.31 vs. 28.71; t = 5.25, p < 10−6 )." I definitively could not see why this information is relevant in the context of the research. The same for "using colons" and "paper length," among others.

"... In contradistinction, some positive attributes of artifacts were actually associated with fewer citations. For example, the mean citations of the 573 papers with a linked artifact, 47, was much lower than the 71.3 mean for the 102 papers with artifacts we found using a Web search (t = −2.02, p = 0.04; W = 22865, p < 10−3247 ). Curiously, the inclusion of a link in the paper, presumably making the artifact more accessible, was associated with fewer citations. This finding strengthens the need for going beyond a quantitative analysis over a single sample established by convenience.

For instance, I did not see some treatment about self-citations.

"Of the 2439 papers, 292 91.8% displayed at some point an accessible link to the full text on GS. What is the impact of these 9.2% over the citations analysis found? Accessibility is definitively relevant in this context.

"These textual relationships may not be very insightful, because of the difficulty to ascribe any causality them, but they can clue the paper’s reader to the possibility of an artifact, even if one is not linked in the paper." I am afraid I could not get the point here.

"As a crude approximation, a simple search for the string “github” in the full-text of all the papers yielded 900 distinct results. Keep in mind, however, that perhaps half of those could be referring to their own artifact rather than another paper’s, and that not all cited github repositories do indeed represent paper artifacts." I could not see how this analysis is relevant to support the discussions.

Additional comments

Figure 1 is not cited. Besides, I understand that this figure could be omitted from the paper.

I suggest combining the analysis presented in Figure 2 and Figure 3 into a single one.

Table 1: please standardize the number of decimal places

2439 papers=> 2,439 papers

"...when skimming these paper*s*..."

The zenodo package is not cited in the paper.

·

Basic reporting

This article is well-written and addresses statistically the very important question if sharing of software artifacts has an effect on citation metrics of computer science, as well as the need for proper archiving of software artefacts for longevity.

The argument is well put forward and supported by a thorough data investigation for the given sample set

Overall language, background and methodology details have been significantly improved in response to peer review.

Experimental design

This research is rigourous and the experimental design is well explained.

The author has responded well to my change suggestions in GitHub and have now added a Zenodo DOI with archive of the data and software.

Validity of the findings

I have not assessed the choices of statistical methods, but the authors have been thorough throughout, e.g. providing p-values and describing method choices. The actual values and figures are also calculated dynamically by the R markdown for the manuscript, and so can be reproducibly verified.

The data set can be still be a bit hard to navigate as it is intermingled with the data, but the strength of that is that the the data results are (for this paper) reproducible. The install instructions have been improved.

Additional comments

Thank you for improving this submission, this is now a very strong contribution.

Repeating from my previous review:

This article is a very welcome addition to the CS field, where sharing of software artefacts have only recently emerged as a practice, catcing up on fields like bioinformatics.

The key message from this article for me is showing evidence for CS authors that sharing artefacts is beneficial for science and for their article's to be cited.

I therefore think this is a very important article for PeerJ CS to publish.

---

## Round 0.3 · Minor Revisions

According to the reviewers, the article has considerably improved in comparison with previous versions. However, they still require a few modifications. Reviewer #1 highlighted the sharing of data and software (should be separated), and also problems in conference acronyms. In addition, Reviewer #1 also requires more explanations in some parts of the article (please see detailed comments). Reviewer #2 highlighted some grammar issues. Please, consider their comments when submitting the new version of the article.

Reviewer 1 ·

Basic reporting

The article is presented in a clear way, with definitions and references useful to the reader. I still would suggest a quick proof-reading to double check for typos and so. For instance, an article seems to be missing in the sentence starting as “An additional goal of is an exploratory”.

First mention of a research artifact (including data, publication, software, etc.) should be accompanied by a reference, even if detailed later (this makes it easier for readers interested in that element to directly go for it rather than looking for the reference somewhere else in the text). This is not the case of the dataset produced together with this study, first mentioned on line 115.

Regarding the sharing of data and software. For any future occasion, I suggest separating data from software as they might have different licenses (unless using a data version for both is indeed intended).

Some conferences might have the same acronym. I suggest expanding the name and adding a link to their websites (if still available).

Experimental design

I suggest making clear from the beginning that the only research artifacts analyzed in this study are in the form of software (or systems software) excluding data, workflows and others.

Step 6 (line 200) needs more detailed explanation. I (randomly) checked the file https://github.com/eitanf/sysconf/blob/master/data/papers/ASPLOS.json from the GitHub repo corresponding to this study, it includes some “citedBy” information. The first paper “cherupalli2017determining” appears with 21 in Google Scholar but that does not seem to coincide with the “citedBy” in that file or to the “outCitations” in the file https://raw.githubusercontent.com/eitanf/sysconf/master/data/s2papers.json. Based on the description on the GitHub repo “papers/: A collection of JSON files, one per conference, with Google Scholar statistics on each paper in the conference”, I was expecting to find 21 “citedBy” items there for the mentioned paper but I saw more. Again, this was a random and quick look but still it might be worth to add some more information on this step (and maybe the others).

Validity of the findings

I find this study of value as we need to understand better how research artifacts beyond the paper itself are shared and what impact they represent in terms of recognition (e.g., in the form of citations) and reproducibility.

One of the limitations of this paper that is not discussed at all (it is maybe outside the scope) is exclusive focus on software as research artifact. Sharing and citing data has been encouraged longer than sharing and citing software. One question that arises is whether the analysis on citation count would change across papers only sharing data, only sharing software or sharing both. One of the premises in this study is that paper sharing software is more reproducible and therefore it would be preferred by researchers, i.e., traducing in more citations. I find it difficult to reproduce a paper with data but not software or vice versa.

Additional comments

Thanks to the author for taking the time to address the reviewers’ comments. I find the current version an improvement regarding the previous one.

Reviewer 2 ·

Basic reporting

I understand the author made considerable improvements in the manuscript, especially those addressing the study's characterization and reproducibility. Therefore, I recommend accepting the paper.

Besides, I invite the author to check the grammar of the following sentences:

“These textual relationships may not be very insightful, because of the difficulty to ascribe any causality them, but they can clue the paper’s reader to the possibility of an artifact, even if one is not linked in the paper.”

“…Using a search function on each document (“pdfgrep”) on each of search terms listed above…”, please check the grammar.

Yet regarding the last sentence, please note that the paragraph where the terms are listed reports that “...several search terms were used to assist in identifying artifacts, *such as* “github,” “gitlab,” “bitbucket,”, “sourceforge,” and “zenodo”….” Here, I am afraid the use of the expression “such as” may not let clear whether all terms applied in the searchers are listed (as expected). It may address a reproducibility issue to fix.

Experimental design

ok

Validity of the findings

ok

---

## Author Rebuttal · Round 0.3

Dear Editor,

I'd like to start by acknowledging the time taken by you and the reviewers to reevaluate this paper. The attached version includes revised and expanded sections (Introduction, Methodology, Discussion) that focused on the three improvements recommended by the editor, namely interpretation of findings, threats to validity, and research questions. I also attempted to address every actionable feedback from Reviewer 2, as detailed in the following point-by-point response to their review.
* * *
> The author does not report systematic steps followed for gathering papers' data. For instance, "....several search terms were used to assist in identifying artifacts, such as "github," "gitlab," "bitbucket,", "sourceforge," and "zenodo" for repositories; variants of "available," "open source," and "download" for links; and variations of "artifact," "reproducibility," and "will release" for indirect references." Besides, it is not clear how the author checked out whether a paper received an award.

**I have now detailed the complete data-collection process in a new subsection of the methodology, which should hopefully clarify how each piece of the dataset was collected.**

> At the beginning of the paper, several research questions are presented without a proper justification. These detailed RQ should be addressed and contextualized in the proper section.

**Upon consideration, I agree with the reviewer that these questions are presented without proper justification, and do not directly support the main research question of this article, which is, what is the quantitative relationship between artifact sharing and citations in computer systems?**
**Instead, these questions are better described as interesting but minor incidental findings that are worth reporting (in my opinion) but not detracting from the main thrust of the paper. I've reworded the introduction accordingly.**

> I understand that the authors followed a partially ad-hoc process for gathering papers' data without double-checking, instead of following a systematic one. If true, the author should explicit these relevant threats to validity. Otherwise, the author should report step-by-step how they reached each data to led the package reproducible.
> I could not find any mention to double-checking on the gathered data. It is relevant at least fo support data gathering through non-systematic steps.

**There was actually significant double-checking and data reconciliation process involved, which is now detailed in the new step-by-step data-collection process subsection. Additionally, a new threats-to-validity was added as requested, and it elaborated the challenge and potential weakness of collecting and verifying artifact sharing information.**

> What was the normality test applied? p-value? Yet about normality, the authros state that the data was normalized. However, after that the author argue that "...omit the 49 papers with zero citations to improve the linear fit with the predictors." Outliers? And what about those with >2000 citations?

**I'm unclear on what confuses the reviewer here. I'll try to response to each question separately.**
**1. As detailed in the statistics subsection, t-test (and p-value) was reported when comparing means, and a signed rank-test (and p-value) when comparing medians.**
**2. I could not find a claim in the paper that the data is normalized. The time-since-publication for citations is normalized by measuring citations exactly 3.5 after publications.**
**3. As for the data being normally-distributed, it is not. The paper claims that the data appears log-normal, and therefore the linear regression uses log(citations) instead of citations as the response variable. This transformation "takes care" of the outliers on the right tail (the highly-cited papers), It cannot be applied to zero values, and the log(p+1) transformation would distort the left-tail results too much, in my opinion. I therefore preferred to omit the 49 zero-values but keep the high outliers. To verify, I recomputed the linear model omitting all papers with 1000+ citations and found only negligible differences in the resulting coefficients or significance.**
**4. For t-tests, a log transformation is not required despite the long tail of the citations distribution because of the size of the sample and the central-limit theorem (the means of the distributions are normally distributed, permitting a t-test).**

> "...It is worth noting, however, that most of the source-code repositories in these
artifacts showed no development activity—commits, forks, or issues—after the publication of their paper, suggesting limited impact for the artifacts alone". I disagree about this example of poor impact. Please note that commits and forks are typically made for source code, which is not necessarily the case of research data.

**The statement is specifically restricted to source-code repositories, not data (the vast majority of the repositories contained source code anyway). However, I replaced the word "impact" with "activity".**

> I miss a discussion of the findings towards causality (which seems to be unfeasible) and more statistical tests (that does not help to understand the original results. Some of the complementary analysis seems to address irrelevant papers' features (at least without proper contextualization). For instance: "Incidentally, papers with released artifacts also tend to incorporate significantly more references themselves (mean: 32.31 vs. 28.71; t = 5.25, p < 10−6 )." I definitively could not see why this information is relevant in the context of the research. The same for "using colons" and "paper length," among others.

**These factors are relevant because some studies have found them to be related or predictive of higher citations, as listed in the related work section.**

> For instance, I did not see some treatment about self-citations.

**The papers in our dataset are cited 85,543 times. It would be impractical to try to download all of these citations and then count self-citations, accounting properly for name disambiguation, since much of this process would be manual. So there is no specific analysis of self-citations in this study.**
**That said, I have no reason to suspect that the self-citation behavior would be different for papers with artifacts released than for those without. If the reviewer has any data or references to suggest otherwise, I would be glad to incorporate it. In the meantime, I added a sentence about it to the threats to validity subsection.**

> "Of the 2439 papers, 292 91.8% displayed at some point an accessible link to the full text on GS. What is the impact of these 9.2% over the citations analysis found? Accessibility is definitively relevant in this context.

**This is first addressed in the following text (closing paragraph of section 3.3). Slow paper accessibility is associated with fewer citations.  In the linear regression, inaccessible papers were assigned "1,000" months time-to-eprint, and this factor was found to be statistically significant but of minimal effect (coefficient) in Table 4.**

> "These textual relationships may not be very insightful, because of the difficulty to ascribe any causality them, but they can clue the paper's reader to the possibility of an artifact, even if one is not linked in the paper." I am afraid I could not get the point here.

**The effort to identify artifacts associated with papers is both manual and error-prone, as elaborate in the paper. This tidbit is offered to other researchers attempting to collect such data, perhaps in order to reproduce these results. Knowing that certain textual properties are associated with a higher chance of locating a related artifact, even if not linked to from the paper, can focus and accelerate independent searches for such artifacts.**

> "As a crude approximation, a simple search for the string "github" in the full-text of all the papers yielded 900 distinct results. Keep in mind, however, that perhaps half of those could be referring to their own artifact rather than another paper's, and that not all cited github repositories do indeed represent paper artifacts." I could not see how this analysis is relevant to support the discussions.

**I've removed this analysis from the paper.**

> Figure 1 is not cited. Besides, I understand that this figure could be omitted from the paper.

**Fig. 1 is cited (2nd paragraph of Sec. 3.1, line 214 in the original revision and now 231). I think it is important to show the wide distribution of sharing rates by conference, and the tendency of ACM conferences to have higher sharing rates, so I left it in place.**

> I suggest combining the analysis presented in Figure 2 and Figure 3 into a single one.

**Both figures show related information, but on different scales. Since there is no space constraint, I prefer to leave them as is than to create a single figure that is harder to interpret.**

> Table 1: please standardize the number of decimal places

**Corrected.**

> 2439 papers=> 2,439 papers

**Corrected throughout.**

> "...when skimming these paper*s*..."

**Corrected.**

> The zenodo package is not cited in the paper.

**I've added the DOI to the existing Zenodo package citation (see Code and data availability section).**
* * *
Sincerely,

Eitan Frachtenberg.

---

## Round 0.4 · Minor Revisions

One very minor point of clarification is required.

Reviewer 1 ·

Basic reporting

No comments

Experimental design

No comments

Validity of the findings

No comments

Additional comments

Thanks for taking into account all the comments raised by reviewers.

Two comments on this new version.

In line 202 “to” is duplicated, see “closest to to the”.

One more comment related to the rebuttal letter.
Comment by reviewer: I suggest making clear from the beginning that the only research artifacts analyzed in this study are in the form of software (or systems software) excluding data, workflows and others.
Response by the author: That is not quite the case. I looked at all the research artifacts I could find in repositories and digital libraries. A minority of those were pure data or configuration files (even when hosted on github.com). They are treated equally to the code artifacts in this study.

The article is titled “Software artifacts and citations in computer systems papers”. The abstract mentions software again “The availability of these software artifacts is critical” and the Introduction section mentions code “an important step towards this goal is the sharing of artifacts associated with the work, including computer code” and then the rest of the article mentions “artifacts” in general. It puzzles me that the title is about software artifacts but the article is about research/experimental artifacts.

---

## Round 0.5 · accepted · Accept

All points raised by the reviewers were addressed. The paper can be accepted!

Reviewer 1 ·

Basic reporting

Thanks for the revised version. I think all points raised by reviewers have been now addressed.

Experimental design

No comment

Validity of the findings

No comment